# Management of a Focal Introduction of ASF Virus in Wild Boar: The Belgian Experience

**DOI:** 10.3390/pathogens12020152

**Published:** 2023-01-17

**Authors:** Alain Licoppe, Valérie De Waele, Céline Malengreaux, Julien Paternostre, Amaury Van Goethem, Daniel Desmecht, Marc Herman, Annick Linden

**Affiliations:** 1Department of Natural and Agricultural Environment Studies, Public Service of Wallonia, 5030 Gembloux, Belgium; 2Fundamental and Applied Research for Animals and Health (FARAH), Faculty of Veterinary Medicine, University of Liège, 4000 Liège, Belgium

**Keywords:** African swine fever, Belgium, control measures, fence, epidemic, wild boar depopulation

## Abstract

African swine fever (ASF) is a fatal disease of suids that was detected in wild boar in Belgium in September 2018. The measures implemented to stop the spread and eliminate the African swine fever virus consisted of creating restriction zones, organising efficient search and removal of carcasses, constructing wire fences, and depopulating wild boar in the area surrounding the infected zone. The ASF management zone included the infected and the white zones and covered 1106 km² from which 7077 wild boar have been removed. A total of 5338 wild boars have been qPCR-tested and 833 have been detected ASF-positive. The search effort amounted to 60,631 h with a main focus on the infected zone (88%). A total of 277 km of fences have been set up. The main cause of mortality in the infected zone was the virus itself, while hunting, trapping, and night shooting were used together to reduce the wild boar density in the surrounding white zones. After continuous dispersion of the virus until March 2019, the epidemic wave stopped, and the last fresh positive case was discovered in August 2019. Hence, Belgium was declared free of the disease in November 2020.

## 1. Introduction

African swine fever (ASF) is a viral haemorrhagic disease with a high case-fatality rate that only affects domestic and wild suids. The disease has been endemic in many sub-Saharan countries since it was first reported in Kenya in 1921 [1]. There have been several incursions in domestic pigs into Western Europe before 1995 which were all successfully eliminated except an endemic situation in Sardinia, Italy [2]. They were all related to ASF genotype I strain. A crucial event was the introduction of the ASF genotype II virus strain in Georgia in 2007 [3] from where it rapidly spread to neighbouring countries [2]. Since then, the disease has circulated widely in domestic pigs and wild boar, endangering the pork industry in Europe and Asia [4,5].

Belgium declared its first cases in wild boar in September 2018 [6]. The virus was detected in southern Belgium (Gaume near the city of Etalle, region of Wallonia), the most western case in Europe at that point. The ASF virus strain identified was genotype II. The complete genomic sequence of the Belgium/Etalle/wb/2018 virus is available on GenBank (#MK543947) and comparative analysis demonstrated homologies with strains circulating in Eastern Europe [7]. This focal introduction is of unknown origin and a judicial investigation is ongoing. The ASF outbreak in wild boar (ASFO-WB) is most likely linked to a human-mediated activity since the nearest infected area in 2018 was, at that time, located in the Czech Republic at about 1000 km distance away. Belgium immediately implemented a series of preventive and control measures, and no case was ever detected in domestic pigs in the country during that time. After 26 months of management, Belgium regained its ASF-free status at European Union (EU) level in November 2020. In December 2020, the World Organisation for Animal Health (WOAH) approved the ‘all swine’ ASF free status of Belgium. 

This paper aims to describe accurately and chronologically the evolution of the epidemic together with the wildlife management measures set up to stop the spread and eradicate the ASF virus. Given the difficulty of reporting a particularly complex spatiotemporal event, the narrative of this paper is based on six periods (over two and a half years) corresponding to six key periods of the ASFO-WB and two distinct geographical areas: where the disease emerged (Gaume) and across which it spread later (Ardenne) six months after the implementation of the strategy against ASF. The data provided should inform more specific research to better understand the virus’ spread under European temperate conditions and to highlight the most efficient measures of control in the context of focal introductions of ASF into wildlife. 

## 2. Materials and Methods

### 2.1. Study Area

The Gaume region corresponds to a sub-region of Wallonia located at the extreme South-East of Belgium against the French and Luxemburg borders (Figure 1). This area was the subject of the first EU disease control zoning for ASF in September 2018. By February 2019, the Belgian Ardenne massif, located just to the north, was also positive for ASF virus. Gaume and Ardenne present a slightly different ecological context (Table 1). A total of 1106 km² was managed against ASF with a maximum proportion of 60% considered as infected. 

From a hunting regulation point of view, the entire ASF management area covers 4 hunting management units encompassing 331 different hunting grounds. The ASF management area concerns 56 forest districts in 5 different entities of the Public Forest Services. The public forests comprise 369 km² (65%) out of the 572 km² of the forested area. From the wild boar’s perspective, the environment is favourable since it is composed of old-grown stands of the most common mass seeding species (*Fagus sylvatica* and *Quercus* species). The proximal agricultural plains are essentially covered with meadows and a few maize and cereal crops (wheat and oats, etc.). Natural sources of water (numerous rivers and springs) are present everywhere and throughout the year. Boar densities are considered average for Wallonia with 3.12 boars shot per km² of forest, based on statistics provided by the hunting councils in 2017 corresponding to the last hunting season before the first detected ASF-positive wild boar. 

### 2.2. Management Measures and Data Collection

As soon as the ASFO-WB was confirmed, control measures were implemented against the spread of ASF in the wild boar population in the Gaume region. These included the establishment of restriction zoning, targeted fencing, active searches for carcasses, and depopulation measures. 

A strategic committee comprising the competent authorities and relevant administrations, as well as veterinarians, epidemiologists, and wildlife biologists was set up, to organise the ASFO-WB management. 

As the implementation of the measures evolved according to the epidemiological status, the results are presented according to six key periods. The key periods were defined as follows:Period 1—From notification to first management zoning (September 2018);Period 2—From the first management zoning to the first extension of the infected zone (October to December 2018);Period 3—The period of successive extensions of the infected zone (January to March 2019);Period 4—The post-epidemic expansion period (April 2019 to December 2019);Period 5—The close of the crisis and the recovery of the ASF-free status (January to November 2020);Period 6—The post-crisis (December 2020 to March 2021).

#### 2.2.1. Restriction Zones

A provisional infected zone was delineated in September 2018 and the first measure was a complete standstill in the infected forest. Halting all activities meant that tourism and forestry activities were forbidden but also that feeding of game species and all forms of hunting were totally prohibited.

Regulated EU disease control zones were imposed in November 2018, as required by the Commission Implementing Decision 2014/709/EU, and then progressively adapted in accordance with the evolution of the epidemiologic situation. The EU zoning included Part II (infected zone with ASF-positive wild boar) and Part I (buffer zone surrounding Part II and in which no case of ASF had been recorded) (Appendix A).

A state operational zoning system was also defined, mapped, and adapted in parallel to the EU one to facilitate response management measures. This zoning distinguished between the area encompassing all the locations where ASF-positive wild boars were detected, the so-called infected zone (ZI), and a buffer area, free from the disease, called the "white zone". The operational zoning was further subdivided to adapt the management strategies to the local epidemiological specificities.

#### 2.2.2. Carcass Search and Removal

The second measure was the organisation of an enhanced passive surveillance through a planned search effort and removal of wild boar carcasses by the State Authorities. Hundreds of people were involved to varying degrees: the regional forestry services (77%), the military (11%), hunters (5%), other government departments (3%), a dog team (1%), and other stakeholders (3%). The search of the carcasses was organised using groups of six to eight people supervised by a local forestry officer. This group size was supposed to be a good compromise between enhanced efficiency (area covered per day) and reduced risk of disturbing wild boar, provided that two adjacent areas were not searched the same day. Even if searching was continuous, thanks to the local forest services, the most important survey operations were organized in an iterative approach based on five-day periods at a rate of once every one to three weeks. The carcass searching intensity was measured through the searching effort, i.e., the time spent for searching and the area covered by the searching teams. Both variables were recorded and mapped from the beginning. Hunters were obliged to report the presence of carcasses to the authorities. Each discovered dead wild boar was located using Global Positioning System (GPS) coordinates, their location and description were then transmitted to the Civil Protection via the WhatsApp (WhatsApp, Inc. ^®^Meta Platforms, Inc., Menlo Park, CA, USA) application to organise its removal by a dedicated team. 

#### 2.2.3. Fencing

The third measure consisted of building fences to counter wild boar movement and to slow down the virus dissemination. The fences were constructed to a standard 1.2 m high wire and unburied, namely the type of fences commonly used to mitigate wild boar damage to agriculture crops in Wallonia. 

Fencing was expected to artificially increase the fragmentation rate of the landscape. They were built along roadsides and took advantage of existing elements acting as physical barriers (i.e., urbanised areas). The length of the fences was measured in the field using a hand-held GPS or remotely on a Geographic Information System (GIS). The dates of starting and ending of fence building were also recorded. 

#### 2.2.4. Depopulation

Depopulation aimed at first to create a so-called white zone free of wild boar around the infected zone. Later, depopulation was implemented inside the infected zone when the number of new ASF-positive cases declined. Basically, the hunters were responsible for depopulation in the white zones with the support of the authorities, while the authorities took complete charge of the culling in the infected zone. In addition to the driven hunts organised by the hunters in the white zones, alternative ways of culling were implemented by the authorities: trapping in collaboration with hunters and night shooting by trained forest officers. Each culled wild boar was identified using a specific traceability tagging and located using exact GPS coordinates (68.9%) or based on the centroid of the hunting ground and an associated report collated by the authorities. The mortality report included information such as date, location, context of mortality, sex, and age. 

Mortality data were documented for every wild boar in the management zone since the beginning of the ASFO-WB in the Gaume area and since December 2018 in the entire management zone (Gaume and Ardenne). 

### 2.3. Biosecurity Procedures and Analyses

The four measures (zoning, carcass removal, fencing, and depopulation) were implemented while fully respecting biosecurity aiming to limit the risk of any human-based dissemination to its minimum. A collection centre was set up inside the infected zone, in an official building of the Authorities (city of Virton). All the wild boars found dead in the infected or white zones were extracted from the field by the authorities with the help of Civil Protection and transported to Virton, as well as all the wild boar culled from the infected zone. All the wild boar culled in the white zones were transported by the hunters to secondary collection centres (n = 2) located outside the infected zone. 

Strict biosecurity measures were consistently followed by the veterinary teams for carcass handling and sampling in the collection centres. An individual post-mortem report was completed including data from the mortality report, as well as body weight, physiological status, carcass integrity, and eventual macroscopic lesions. Targeted organs (spleen on fresh and bones on decaying carcasses) were transported the same day by a specific shuttle to the National Reference Laboratory (Sciensano) in Brussels for virological analysis. Quantitative PCR (qPCR) was performed according to the method described by Tignon et al. [8], amplifying a 159 bp amplicon of the p72 ASF virus gene and a 114 bp amplicon of the swine beta-actin gene as internal quality control. Depending on the context of discovery or suggestive lesions, some cases were prioritised by the veterinary team for urgent analysis. All carcasses (found dead or killed), wherever they originated from in the management zone, were disposed to the rendering plant by a dedicated truck.

Different standard operating procedures were designed and disseminated for every management activity. Biosafety training was organised for the carcass search teams, hunters, fence builders, and forestry workers. People involved in the search of carcasses wore specific equipment and were systematically disinfected at the end of the day. Hunters had been trained for the disinfection of their equipment and for safely packing and transporting every killed boar to the collection centres. They were compensated with 100€ or 50€ (according to the zone) per wild boar. Workers of the fencing companies were also disinfected at the end of their working days before leaving. Furthermore, later on, once logging activities were progressively resumed after 15 months, a company was contracted by the authorities to disinfect the trucks and machines. 

In order to visualise the progression of the disease, convex polygons encompassing the locations of the ASF-positive boars on a monthly basis were generated to create infected polygons. All distances and directions were measured between the outer boundaries of the infected polygons and the newly detected ASF-positive cases. The rate of spread (km/month) was then calculated on the basis of the distances between the positive cases in month n+1 and the boundaries of the infected polygon in month n. This rate of spread was analysed according to time in months and the direction of spread (cardinal points).

To visualise the effectiveness of the fences against virus dispersal, the 10 fence segments closest to the infected boars were visually selected. The ASF-positive cases were then distributed according to their distance from these segments within a buffer of 3 km on either side of the fences. 

The maps were created on ArcMap 10.5.1 (© ESRI), the infected polygons were generated with the Convex Hull tool, the distances and directions with the Near tool, the tables with Excel (© Microsoft), and the graphs with R. 

## 3. Results

### 3.1. Epidemic Pattern

Data from passive and active surveillance revealed the spatiotemporal distribution of the virus during the epidemic (epidemic curves and infected polygons). In total, 5338 boars were tested between September 2018 and March 2021, of which 833 (~16%) tested positive for ASF (Table 2). The infected polygon expanded steadily in forested areas from east to west until January 2019, then north and south in February 2019 (Figure 2). Its expansion stopped in June 2019. The peak of positive cases occurred in February 2019, the last "fresh" positive case was detected in August 2019 in a culled animal, and the very last positive case (a skeleton) was detected in March 2020 (Figure 3). The postmortem interval of ASF-positive bones found in 2020 was estimated to be several months, based on the date of discovery, climatic conditions, and macroscopic appearance (desiccation, degree of bone wasting, presence or absence of tendons, ligaments, and cartilage laxity of intervertebral joints). In qPCR, these bone samples were found to be weakly positive for the intended viral target (Ct > 34) while being concomitantly negative for beta-actin (>45). Additional tests were performed by the WOAH Reference Laboratory (INIA-Madrid), they revealed the absence of infectious viral particles in isolation and hemadsorption assays on PBM cells. These data suggest a cessation of the virus circulation from autumn 2019 onward when the density of the wild boar species in the area had become much lower.

### 3.2. Control Measures

#### 3.2.1. Restriction Zones

The EU and operational zonings did not differ in their respective overall areas (1106 km²) and external boundaries. They differed only slightly in their compartmentalisation between white and infected areas. Using the previously defined time steps, the evolution of the different zones is presented in Table 3. The spatiotemporal pattern of their expansion, the erection of fences and of areas searched for carcasses are illustrated in the six panels of Figure 4. With respect to the infected areas, separate epidemic curves were constructed for Gaume and Ardenne, the underlying rationale being that the former had been colonised by the virus six months before the latter and that, as a consequence, wild boar densities were very different on both sides at the time of virus introduction (densities had already been reduced in Ardenne due to depopulation measures already implemented). A series of successive peaks can be seen in Gaume, reflecting the spatiotemporal evolution of the epidemic in the area, with a maximum in February 2019 and a gradual decline in the following months. In Ardenne, the epidemic did not evolve in successive waves (Figure 5). As the vast majority of ASF-positive wild boars were detected in wooded areas, measures restricting human activities were imposed in these areas and not elsewhere. A detailed description of these measures is available in Table 4, they correspond to the recommendations of the EU Veterinary Emergency Team (EUVet) experts and to the requests produced by the Scientific Committee of the Federal Agency (SciCom).

#### 3.2.2. Carcass Search and Removal

The intensity and spatio-temporal pattern of the search efforts for wild boar carcasses in the field illustrates the general organisation of the operations conducted in the white and infected areas. Search efforts were concentrated in the infected area, with 68% of the effort in the Gaume, 19% in Ardenne massif and this for a total of 53,000 man*hour. The remaining 13% (~8000 man-hours) was dedicated to the white zones (Table 5). Chronologically speaking, search intensity was highest during periods one to three, variable according to the season in period four (less in summer because of the visual screen formed by vegetation), and progressively decreased in periods five and six (Figure 6). Only the forested areas were searched, nine times successively in Gaume, and five times in the Ardenne massif. One man-hour unit corresponded on average to an area of 5.3 hectares covered by a forest officer. On average, each forest officer covered around 20 hectares per working day, with an effective search time of 4 h/day.

#### 3.2.3. Fencing

Within the management area, 237 km of new fences were erected, complementing the 70 km of pre-existing fencing that flanked the nearby highway and resulting in a density of approximately 270 m of fence per km². An additional 40 km were constructed outside the management area (Table 2). After connecting to fences erected in France (132 km) and Luxembourg (10 km), the complete network created 20 enclosures (Figure 7). Initially, the fences were constructed as an emergency measure in the hope of confining boar circulation to the infected area and, hence, of limiting the spread of the virus. That initial network was then centrifugally extended each time or even before (“Somethonne”, “road N801”, and “Suxy”) an animal tested positive outside the area previously held to encompass all positive cases. The construction effort was maintained even after the epidemic peak was reached to reinforce containment in case some infected animals moving outside the infected area went unnoticed and to facilitate depopulation operations around the infected area.

In order to measure the effect of fencing on the spatial spread of the virus, the 10 fence segments closest to the positive cases were selected (Figure 7). For each, the distribution of distances from the discovery/culling site of each positive case was then determined (Figure 8). In total, 6 of the 10 segments directly exposed to the spatial expansion front of the virus successfully contained the virus. For example, the “Orval” segment, erected in the middle of a forest along a public road, dramatically stopped the epidemic wave moving from east to west in the Gaume forest. The segments “highway E411-Stockem”, “Messancy”, and "Musson", erected in the most urbanized areas, were not crossed, although positive cases were found close to the fence. Finally, the "Mellier" and "Les Fossés" segments have apparently never been crossed either. Conversely, four fence segments were found to be porous. In January 2019, the "Somethonne" segment did not prevent the centrifugal movement of a herd of wild boar (some of which were ASFV positive) towards a still untouched agricultural plain. The "N891" segment had been erected along the national road N891 in an area characterized by large open agricultural areas, sparse and patchy distributed wooded areas, and more numerous gates due to scattered urbanisation and agricultural activities. A positive case located 10 km north of it (a carcass still in good condition) was detected in February 2019. Regarding the road "N801" segment, a virus-positive wild boar leg was discovered outside in March 2019. However, there is still some doubt as to whether this segment was indeed constructed outside the infected area. Despite the active search, it could be that this area was already infected before building this segment. Finally, a virologically positive, adult, male boar was trapped outside the "Suxy" segment in June 2019. Despite the fact that an absolute seal was not observed, which could be assumed from the outset, our observations show that the spatial expansion of the infected polygon was powerfully impeded by the erection of the network of fences.

The median rate of spread (Figure 2) was lower than 2.0 km/month on average between emergences (October 2018) and February 2019 (Figure 9). At this time, a 10 km jump northward occurred by crossing the road "N891", which extended the infected area to the Ardenne massif (Figure 4, panel 3) and required new containment measures. Concomitantly, an escape occurred to the south, with a jump of about 4 km and a corresponding expansion of the infected area, this time without consequence on the management zoning. After February 2019, the number of virus-positive animals found outside the infected polygon and the distance these animals crossed beyond the fences drastically decreased until a final stop of the expansion in the summer of 2019. The effect of the fence network on decelerating the spatial spread of the virus was amplified by the collapse of wild boar densities, both inside the infected area due to the mortality rate associated with the infection, and outside due to depopulation operations. 

#### 3.2.4. Depopulation

Quantitative data on the temporal reduction of wild boar densities are presented in Table 2. In the first infected area (Gaume), a very large excess of mortality was observed at the time of arrival and initial spread of the virus, with the discovery of 1181 carcasses of which 801 tested positive for ASF. In the second infected massif (Ardenne), densities had been drastically reduced by culling and trapping operations before the virus entered the massif. Subsequently, the infected area was emptied of the remaining wild boar thanks to successive night shooting campaigns on baiting sites fitted with remote warning camera systems. In the white zones, culling consisted of driven hunting, trapping, and night shooting. The highest cull rates were achieved in 2019 (periods 3 and 4) thanks to a strong focus on trapping. When densities decreased (periods 5 and 6), night shooting became the most productive way of culling (Figure 10). At the end of period 6, an estimate of the minimum number of wild boar still present in the infected area was made based on camera traps on baiting sites, visual sightings during hunting, and night shooting sessions. In the infected zone, we estimated the number of wild boar at 59, i.e., 0.10/km² or 0.22/km² of forested area. In the white zone, the remaining population was estimated at 156 animals, i.e., 0.31/km² or 0.61/km² of forested area, but with a great heterogeneity between sectors.

Comparing the mortality data collected during the crisis (2018–2021) to the hunting mortality data from the previous hunting season (2017) highlights that the mortality rate observed in the infected zone doubled over the first year of the ASFO-WB and then decreased drastically until 2021. In the white zones, the mortality sharply increased during the two years following the ASFO-WB and then decreased slightly. These data are put into perspective with the statistics for the whole region of Wallonia (Table 6).

## 4. Discussion

On September 2018, the presence of ASF was confirmed in wild boar in Belgium. This was the first time since an outbreak in pig holdings in 1985 [9]. As a preventive action at the Federal level, all the pigs in the initial infected area were immediately culled and intensive biosecurity measures were implemented across the whole country to prevent the transmission of the virus into pig farms. The main objective was the elimination of the disease and specific control measures were deployed in wild boar populations according to the only successful EU experience from the Czech Republic [10], the EUVet experts’ reports in Belgium, and the European Food Safety Authority (EFSA) recommendations [11].

### 4.1. Zoning

The size of the infected zone (corresponding approximately to the Part II set by the Commission Implementing Decision 2014/709) is quite large compared to the entire Belgian ASF management zone (Parts II and I). A relatively smaller-sized white zone (Part I) facilitates a more intensive and therefore more effective implementation of management measures, in particular depopulation. To this small-sized Belgian white zone, those from France and Luxemburg (299.1 and 3.6 km², respectively) must be added. Both countries collaborated intensively with Belgium in applying the same coordinated measures, in a comprehensive way, in terms of e.a. depopulating, fencing, and searching for carcasses [12]. Given that the location of the Belgian ASFO-WB is very close to France and Luxemburg, the transboundary collaboration distributed the efforts among three different countries. An ASFO-WB in the very centre of Wallonia had probably led to the delineation of wider white zones (EC Zone I) by the EU Authorities with a probable dilution of the management efforts leading to a lower efficacy.

At the time the virus was detected, approximately 125 km² of forest were declared infected (see infected polygon of September 2018 in Figure 3). Compared to the situation in Zlin, Czech Republic, this corresponds to an area three times larger [13]. This later detection is partly related to the fact that the ASFO-WB developed initially far from regularly visited areas. That is why the standstill focused on the forest because the first cases were detected in the heart of a large woodland and the main type of adjacent agriculture consisted of meadows which are not an attractive resting place for wild boar compared to other types of high-cover crops. The only local maize fields were already harvested at the time of the first detection. Again, compared to the Czech situation, even if similar, the R_0_ value was lower in Belgium [14], suggesting a lower density of wild boar. While tourism and harvesting activities were forbidden in the forest, hunting was forbidden in the entire infected zone. Fortunately, the first cases were detected before the beginning of the hunting season which would probably have dispersed the virus more quickly [15] by increasing the contact rates between wild boar populations and therefore driving virus transmission [16]. The spread of the virus followed the natural forested corridor of the Gaume’s forest from east to west at a near predictable speed of about 1.56 km/month during autumn [17]. Unexpected jumps occurred during winter through the countryside despite the fence network. Lightly wooded agricultural areas cannot be considered as obstacles, at least in certain seasons. This is surprising since Podgórski and Śmietanka [18] found a very regular pace of spreading of 1.5 km/month in North-East Poland in a mosaic of field and forest, a comparable landscape to Gaume. They did not find any connection between the main biological requirements of wild boar and the speed of the ASF spread. Human intervention or delayed detection could also explain such unpredictable jumps.

### 4.2. Carcass Searching

Carcass search and proper removal from the environment increase the probability of successfully eliminating the virus [11,15]. Subsequently, the systematic analysis of the carcasses’ locations allow the approximate delineation of the truly infected zone and tracking of the epidemic phase. In the Belgian model, the reactivity of the testing process (from sampling at the collection centres, testing at the NRL, and transmission of results within 24 h) was crucial. This allowed prompt front-line decisions to be made (e.g., building new fences) based on the location of new positive cases.

The way of selecting the search areas necessarily evolved according to the phase of the epidemic. During the epidemic period (September 2018–March 2019), carcass search was directed towards the hot spots or on the edge of the infected areas. In the post-epidemic period (April 2019–March 2020), the objective was to ensure the best possible coverage of the infected zone and direct surroundings. In both cases, a transversal monitoring system of the searching effort and the covered areas must be set up within a single GIS to define priorities for the following weeks, but also to avoid leaving areas uncovered. All data must then be centrally managed by a single authority to ensure a coherent organisation of the carcass search. The carcass detection model [19] was useful to establish priorities but, based on the 200 first positive cases detected in autumn 2018, it probably requires further improvement to take account of seasonality.

Enhanced passive surveillance as organised in Belgium was insufficient to detect all cases of ASF-positive boars outside the infected area. Crossings through the ASF fences were identified at least four times. In only one of them, organised searching for carcasses allowed the discovery of a new infected zone. The other ASF-positive cases had been noticed thanks to active surveillance (culled animals during hunting and trapping) or by chance, thanks to the awareness of hikers (e.g., in the case of the 10 km jump to the North in February 2019).

Searching focused mainly in woodlands where the probability of detecting ASF-positive carcasses is highest [20]: we found 702/801 (88%) ASF-positive carcasses in forested habitat as in the Czech Republic where a majority of wild boar were detected in forests even though its proportion was lower [21].

The implementation of the searching for carcasses was based on the availability of professionals, mainly forest officers from the regional administration. During the epidemic phase, occasional reinforcements from the army made it possible to cover large areas, but their implementation was not easy due to a lack of field competence. As far as private landowners, hunters, or other volunteers were concerned, their support varied greatly among hunting grounds. Relying only on them was not sustainable over time, especially since financial support for carcass searching was not available. Later, the local foresters were highly supported by the hiring of staff especially dedicated to this task, but this manpower support arrived very late in the process. Unmanned aerial vehicles were not used, except on one occasion over swampy areas. A team of sniffer dogs gave encouraging results. Unfortunately, they could not be tested during the epidemic period because they were not trained in time. In any case, they should be considered as a complement to help and not as a full solution. Several trained teams of sniffing dogs would be necessary for an intensive search and maintaining their training outside of a crisis period is challenging. An important element to ensure that sufficient searching efforts could be provided is the availability of large number of professionals, who can be mobilised sometimes at very short notice. Maintaining motivation over the long term was complicated even with professionals, especially when no further carcasses were being found. In the end, the time spent searching was around 90% for regional (79%) and national (11%) authorities. The recorded effort spent searching by hunters was lower than 10%, for the reasons stated above.

### 4.3. Fences

There was no evidence that fencing significantly contributed to stopping the spread of ASF in wildlife because it is not possible to have a 100% proof wild boar barrier, particularly when the landscape is crossed by several streams [22]. In fact, the effectiveness of fencing in disease management, especially among wildlife, has not been well documented. The implementation of fencing as a tool to mitigate disease spread retains a large degree of uncertainty [23]: both practical and biological details have to be considered. For example, wild boar fences should be conventionally buried at a 50 cm depth to attain satisfactory efficacy. Maintenance of fences is always key: they must be periodically inspected and repaired [13]. Nevertheless, the construction of non-buried wire fences was widely used in the Belgian context. The approach used seemed to be a good compromise between efficacy, compared to electric fences, and rapid installation, compared to buried wire fences. Placing fences along the state roads avoided issues of ownership and also profited from the already existing barrier effect of the public roads. Fences were also used to materially demarcate some management zoning with an intended effect on public awareness. Fences appeared to be breached at least four times, most of them during winter. It appears that fences along urbanized zones or fenced highways offered a good barrier against the boar’s spread. The most problematic instance of crossing fences corresponded to a mosaic landscape of mixing small forests and pastures with disrupted urbanization and many gates.

Together with French and Luxemburg fences, 20 enclosed areas were created with areas ranging from 17 to 418 km², i.e., corresponding to 20 wild boar populations to be managed according to their epidemiological status. In the white zone, where depopulation was the priority, avoiding any risk of migration greatly facilitated the culling effort. Without knowing what the optimal area to fence is, some recommendations can be made. It should not be too small (>50 km²) to fulfil all the requirements of a boar population hence limiting the risk of evasion, and not too large (<200 km²) to allow intensive culling measures and efficient maintenance of the fences. The creation of a white zone around the infected zone is the best guarantee to prevent any circulation of the virus thanks to the elimination of the virus naïve hosts. Fencing off the area to be cleared from wild boar is a prerequisite to prevent any population immigration from outside or any exchange with the infected zone. Applying culling pressure in an unclosed area would be futile in the context of a continuum of forested landscapes and high wild boar densities. The results obtained in Belgium, demonstrating a drastic decrease in the population, would have been unlikely without a pre-containment of the population.

### 4.4. Depopulation

The term "depopulation" is borrowed from livestock epidemic management, which involves culling an entire herd on a farm. In wildlife management, achieving a depopulation target is almost impossible. Nevertheless, the term has been widely used to give hunters and other managers a clear objective when the intended final result is a drastic reduction in numbers.

The delimitation of the white zones is challenging, and the wild boar reduction measures must be completed in the short term, i.e., within a few months [24]. Very little guidance exists about the significant and short-term reduction of the wild boar population. EFSA recommendations underline the importance of combining different ways of culling to achieve a quick and drastic reduction in wild boar population [22]. In the Belgian experience, even if highly variable, the role of citizen hunters in the depopulation effort was significant in that they organised more driven hunts than in previous years. In general, the closer the nearest ASF-positive wild boar, the higher the participation rate by the hunters in depopulation. However, hunting alone would probably not have been enough to achieve the depopulation objectives [25], not only from a technical point of view, but also because of the lack of motivation since hunting is primarily a leisure activity and not a public service [26]. It also appeared that local hunters were more inclined to participate in the depopulation than those coming from elsewhere to hunt occasionally in Gaume. The addition of complementary measures implemented and supervised by the regional administration, such as intensive trapping and the organisation of night shooting on the agriculture fields or at baiting points in the forest, was essential. Yet, these measures had to be intelligently arranged in space and time to optimise their effectiveness and their acceptance. Driven hunts are efficient when organised outside the vegetation growing season in autumn and winter, with trapping taking over in spring and summer [27]. Night shooting can be practiced all year round without any interference. Nevertheless, it took a lot of negotiation with the hunters to let the authorities shoot wild boar in the white zone. As reported by Urner et al. [28], some ASF control measures, conducted by the authorities, are perceived as unpleasant and unethical by the hunters.

During the epidemic phase, the virus kills much more efficiently than any kind of culling [29]. Moreover, driven hunts are supposed to facilitate the spread of diseases since they induce the movements of a large proportion of the disturbed population [30]. Letting the disease devastate the wild boar population without any hunting operation reduces both the population down to circa 80% [29] and the consecutive risk of entering an endemic phase [15]. Moreover, leaving the area free of any hunting activity also reduces the risk of a human-associated long-distance spread of the virus [31]. In the infected zones, once devastated by the disease, the shooting of the final individuals was only possible thanks to camera traps equipped with remote transmission systems, which greatly facilitated the intervention of the night shooting teams. Modern technology such as infra-red spotters and scopes to detect wild boar and shoot them safely is also essential to increase the effectiveness of these culling operations.

Both fencing and depopulating have played a significant role in the extinction rate of wild boar according to a contemporary camera trap-based study aiming to compare the occupancy of wild boar in time and space in a representative part of the ASF management zone from March 2019 till June 2020 [32]. 

### 4.5. Challenges and Perspectives

In spite of the success in tackling ASF from wild boar, the economic consequences in the infected area should be acknowledged. A non-exhaustive list of collateral damages includes the temporary disappearance of all pig farming; the suspension of forestry activities such as plantations or their maintenance; the removal of harvested timber or firewood; the postponement of exploitation works; complication of the bark beetle crisis management; impact on the local citizen; leisure pursuits and green tourism in general; loss of value of the hunting grounds; responsibility for damage to crops due to the prohibition of hunting; and increased pressure from wild herbivores (roe and red deer, mouflon) on forest regeneration after two years without hunting in some areas.

This explains the perceptible tensions locally, between the administration and the landowners; the administration and the hunters; and the local and central administrations, while the federal and international authorities held up the management of the crisis as an exemplar model. As in any crisis, the human dimension is a key element in the process that should be better taken into an account. A better bi-directional (bottom-up and top-down) communication to all the stakeholders involved in the crisis is probably the key to enhancing the compliance of the measures as suggested by the EFSA Panel on Animal Health and Welfare [33].

The control measures put in place in the context of a focal introduction of the ASF virus in Belgium made it possible to avoid any outbreak in pig farms and to limit the spread of the disease to wildlife as a first step, and then to eliminate it in a second step, over a period of 26 months. These implemented measures seem inseparable from each other as already highlighted for depopulation and passive surveillance by Gervasi and Guberti [34]. The Belgian experience tends to demonstrate the key role of wire fences as a break on the spread of the virus in a mainly wooded and mostly unfragmented context [17]. Successful management also relies on multiple depopulation methods in addition to those usually applied by hunters, e.g., trapping and night shooting in a comprehensive way. The support of an administration that is spatially organised in the field and a scientific council based on both epidemiological and wildlife management skills is an essential element. A more detailed cost/effectiveness analysis of the ASFO-WB results will undoubtedly highlight the relevance of one measure in relation to another. The current estimate of the measures implemented only for wild boar management is around 300 euros/ha. It underlines the importance of effective and continuous passive surveillance in peacetime to detect disease outbreaks in wildlife as early as possible and thus avoid later risks and costs of managing large, infected areas.

## Figures and Tables

**Figure 1 pathogens-12-00152-f001:**
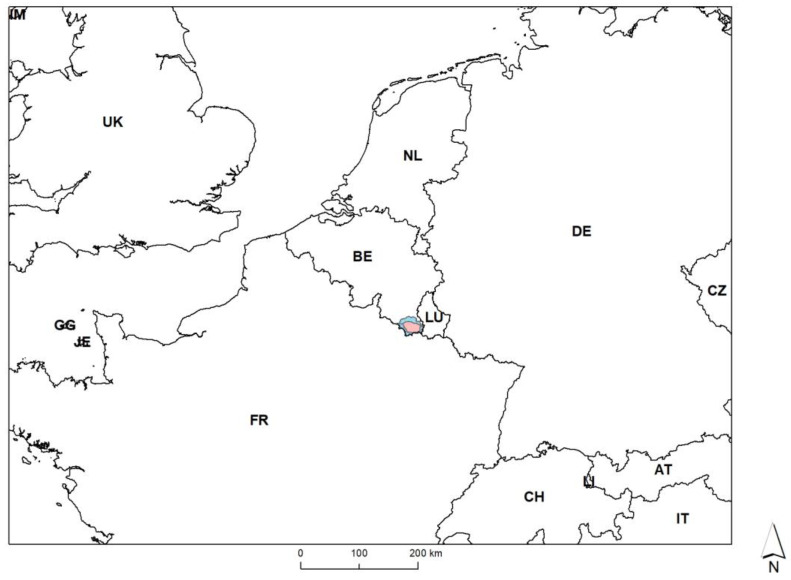
Map showing the Belgian ASF zoning (EU Parts I and II, 27 November 2018).

**Figure 2 pathogens-12-00152-f002:**
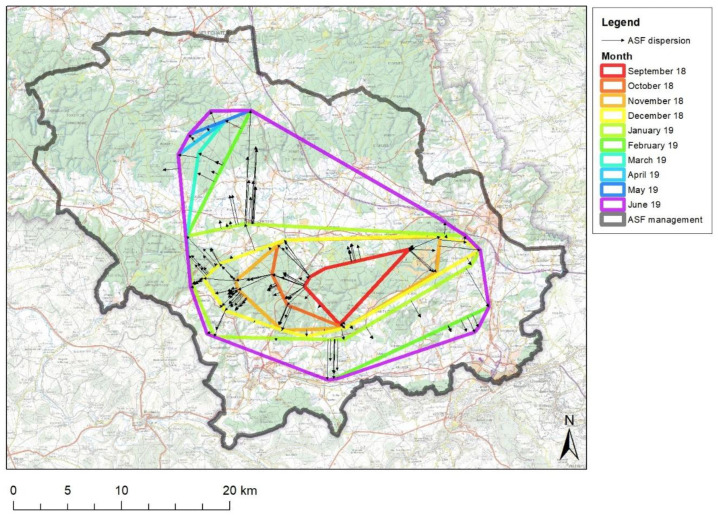
ASF infected polygons were derived on a monthly basis. The vectors show the distance and direction between each outer ASF-positive wild boar of the month from the external limit of the infected polygon of the previous month.

**Figure 3 pathogens-12-00152-f003:**
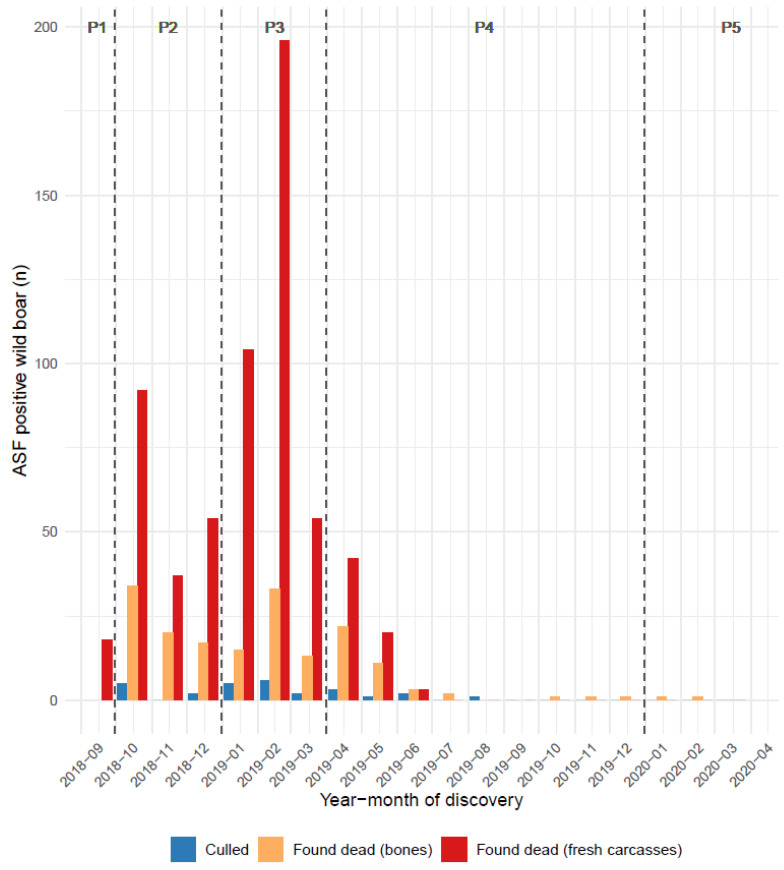
Evolution of the number of ASF-positive cases in wild boar according to time (month and period) with a distinction between those actively culled (blue bars) and those discovered dead (fresh carcasses and bones, red and orange bars) shown.

**Figure 4 pathogens-12-00152-f004:**
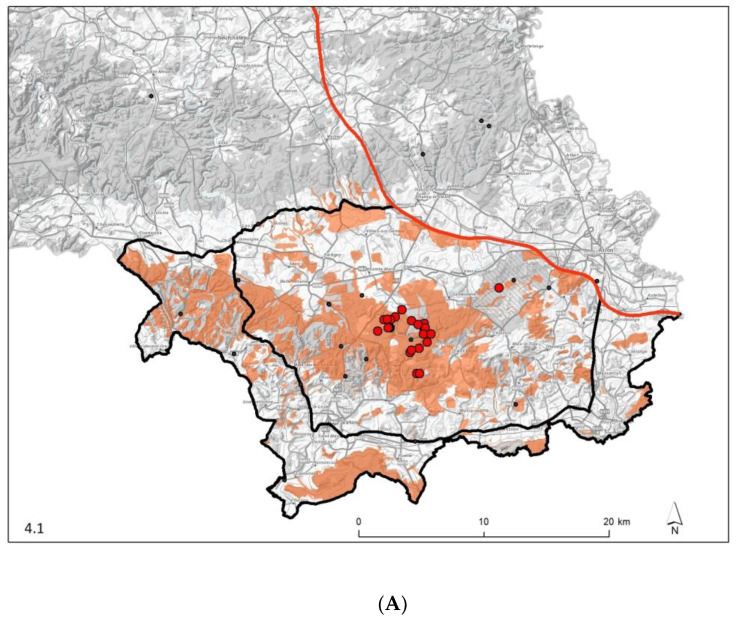
Management zoning tested wild boar (black circles) and positive within them (red circles), area searched for carcasses covered at least once during the period (orange polygons), ASF-fence (purple line) and highway fence (red line) according to the six periods of management: September 2018 (**A**), October–December 2018 (**B**), January–March 2019 (**C**), April–December 2019 (**D**), January–November 2020 (**E**) and December 2020–March 2021 (**F**). (© IGN-NGI).

**Figure 5 pathogens-12-00152-f005:**
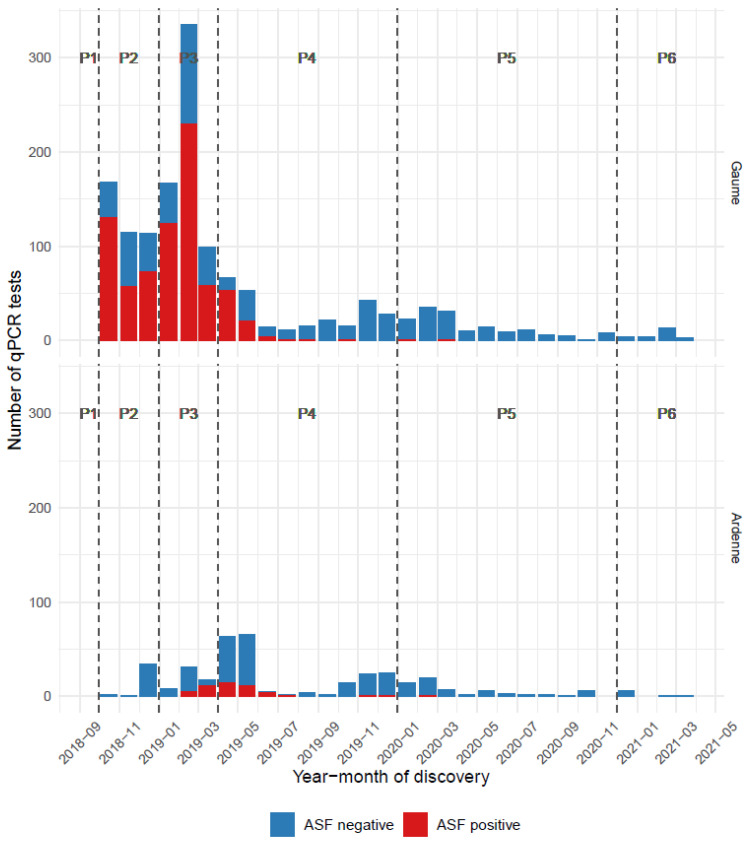
Monthly ASF-qPCR results in wild boar in the infected zones of Gaume (above) and Ardenne (below) from the first detected case in September 2018 until March 2021. This refers to the infected zone at its maximum size.

**Figure 6 pathogens-12-00152-f006:**
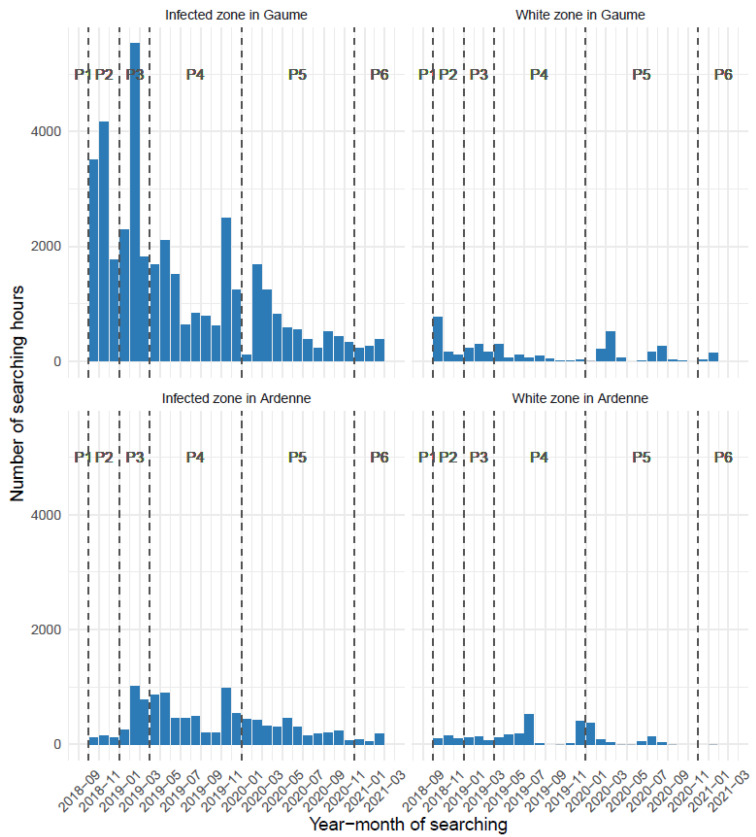
Monthly searching effort (hours) spent to detect wild boar carcasses according to epidemiological status (infected vs. white) and geographical area (Gaume vs. Ardenne) from September 2018 until March 2021 (six periods).

**Figure 7 pathogens-12-00152-f007:**
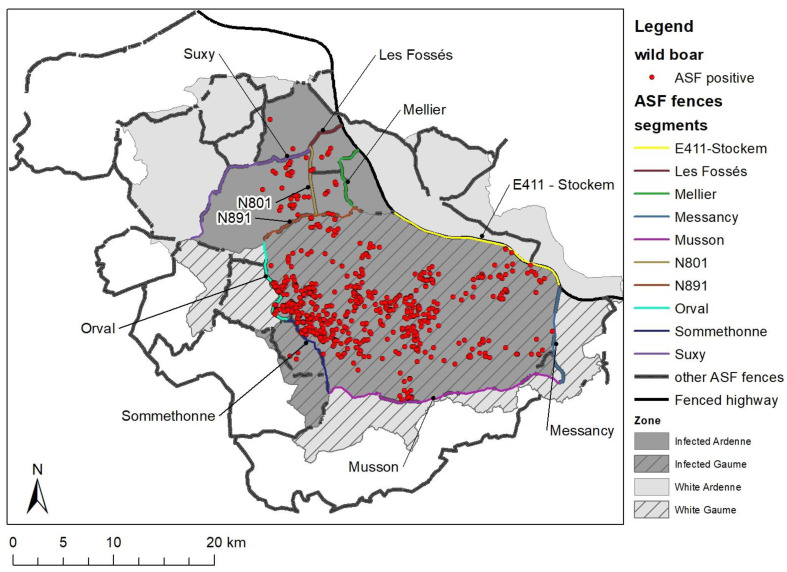
Network of fences in Belgium, France, and Luxemburg, and identification of the 10 fenced segments directly exposed to ASF-positive wild boar.

**Figure 8 pathogens-12-00152-f008:**
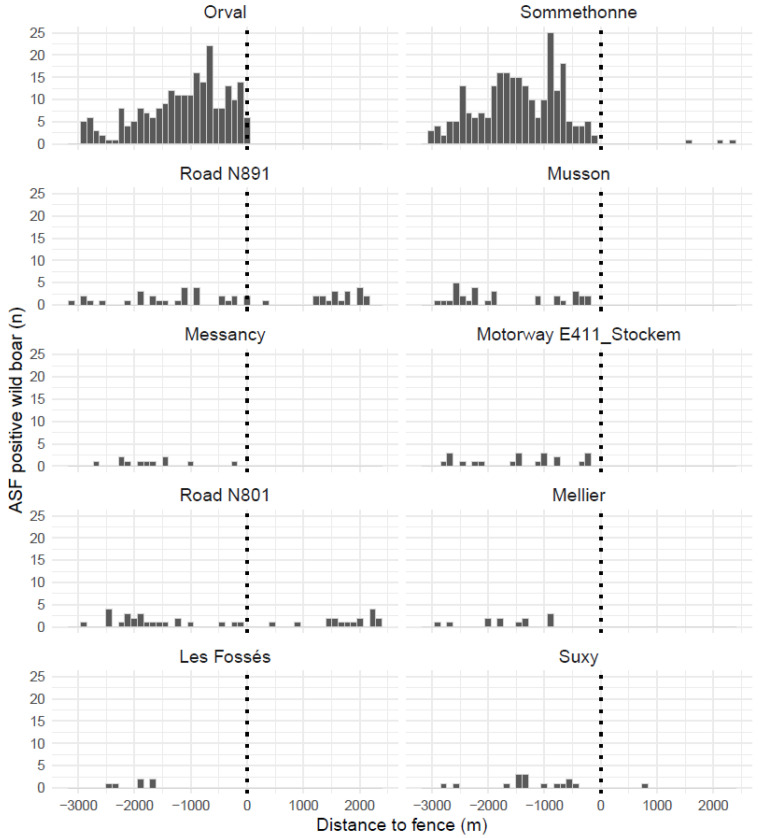
Distribution of ASF-positive wild boar on both sides (infected vs. white) of 10 fence segments facing the expansion of the epidemic as revealed by ASF surveillance. The dotted lines symbolize each fence segment, negative distances are inside the infected area and positive ones are outside.

**Figure 9 pathogens-12-00152-f009:**
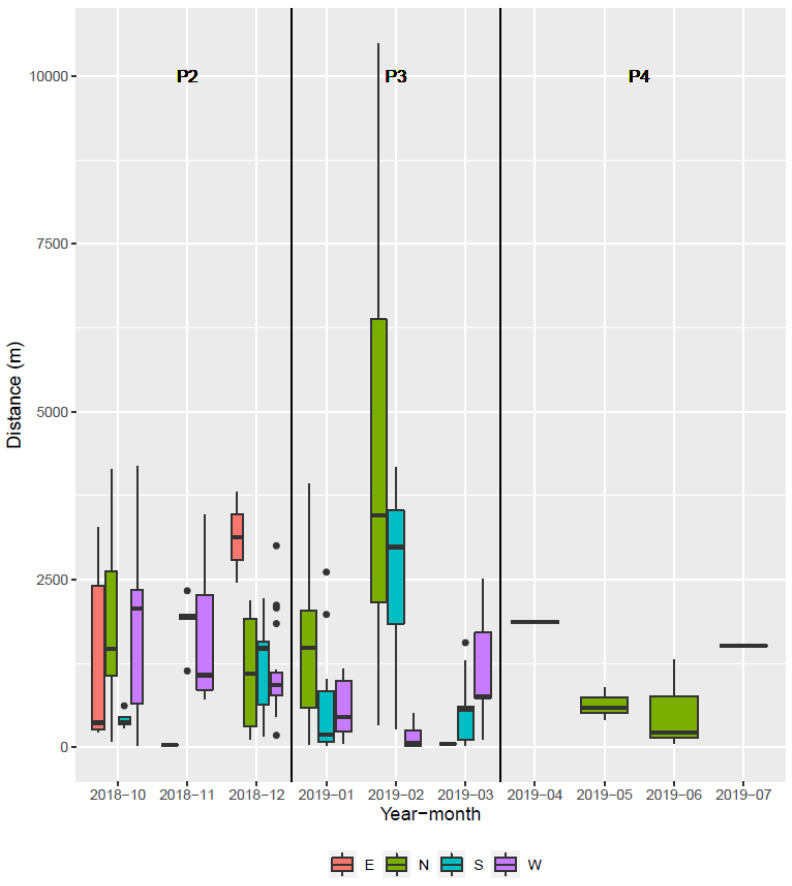
Distribution of the distances of outer ASF-positive wild boar of a given month from the infected polygon of the previous month according to time (month) and direction (cardinal points).

**Figure 10 pathogens-12-00152-f010:**
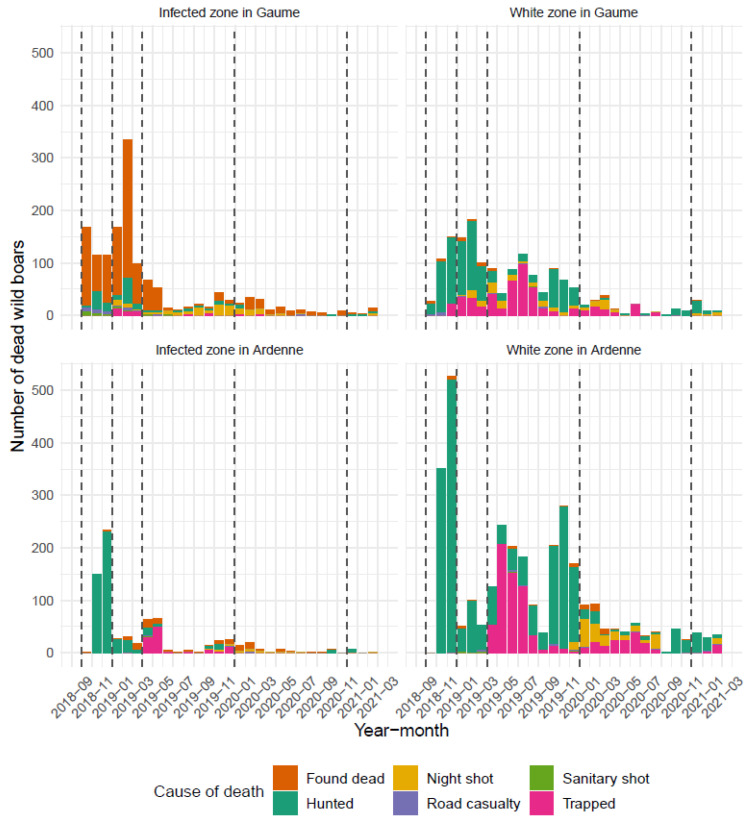
Context of mortality of wild boar on a monthly basis in Gaume and Ardenne according to epidemiological status from September 2018 until March 2021. Dotted lines indicate period changes (six periods).

**Table 1 pathogens-12-00152-t001:** Environmental description of the ASF management zone with a distinction between the initially infected zone “Gaume” and the “Ardenne” zone infected in February 2019 (© IGN-NGI, © Service Public de Wallonie, Lifewatch-Ecotope 2018).

ASF Management Zone 1106 km²	Initially Affected Zone (Gaume) 628 km²	Extension to the North (Ardenne) 478 km²
Forested Area	299 km² (76% deciduous; 11% coniferous; 3% mixed)	273 km² (55% deciduous; 32% coniferous; 3% mixed; 10% open and clearcuts)
Agricultural Area	302 km² (79% pastures; 7.3% maize; 8.8% cereals)	192 km² (85% pastures; 5.4% maize; 6.6% cereals)
Average Elevation	313 m (184–465)	391 m (249–518)
365-Day Climate	P = 966 mm; T = 8.94 °C	P = 1069 mm; T = 8.43 °C
Hunting Bags in 2017	954 (1.51/km² or 3.19/forested km²)	842 (1.76/km² or 3.08/forested km²)

**Table 2 pathogens-12-00152-t002:** Summarised information inside the Belgian ASF management zone (1106 km²) from September 2018 until March 2021, subdivided into 2.1 Gaume and 2.2 Ardenne. According to the periods one to six: length of the ASF and highway fences, surface covered by and hours dedicated to the active search of carcasses, number of dead wild boar according to the context of death, number of qPCR tested wild boar (number of positive for ASF within them), and incidence of the disease. The results about dead wild boar are subdivided into the infected area and white area, at the time when the infected area was maximum.

**2.1 Gaume ** **628 km²**	**18 September**	**October–** **18 December**	**January–** **19 March**	**April–** **19 December**	**20 January–** **20 November**	**20 December–** **21 March**	
Period	1	2	3	4	5	6	Total
N Days	17	91	89	274	334	120	925
ASF Fences (km)		33.2	55	28.2			116.4
Fenced Highway (km)	23.7						23.7
Infected Zone (454 km²)							
Area Searched (km²)	218.6	653.0	369.6	431.3	268.4	57.1	1998.1
Time for Searching (man × hour)	1628	10,182	10,390	11,723	6864	1097	41,883
Found Dead	33 (26)	311 (253)	473 (397)	146 (76)	102 (2)	15 (0)	1080 (754)
Sanitary Shots	3 (2)	11 (7)	4 (4)	3 (3)	1 (0)	0	22 (16)
Road-Rail Casualty	8 (0)	26 (1)	10 (2)	2 (0)	2 (0)	0	48 (3)
Hunted	1 (0)	49 (0)	62 (6)	19 (1)	6 (0)	4 (0)	141 (7)
Trapped	0	0	27 (2)	4 (0)	2 (0)	0	33 (2)
Shot at Night	0	0	24 (1)	96 (1)	41 (0)	6 (0)	167 (2)
Total	45	397	601	271	154	25	1493
Total Tested	45 (28)	397 (261)	601 (412)	271 (81)	154 (2)	25 (0)	1493 (784)
Incidence	0.62	0.65	0.69	0.30	0.01	0.00	0.53
White Zone (174 km²)							
Area Searched (km²)	75.1	134.0	257.0	29.7	48.5	11.0	555.3
Time for Searching (man × hour)	370	1158	1071	615	1221	201	4637
Found Dead	1	11	20	7	11	2	52
Sanitary Shots	1	0	0	1	0	0	2
Road-Rail Casualty	1	12	2	7	3	1	26
Hunted	0	238	294	250	28	28	838
Trapped	0	22	82	299	63	0	466
Shot at Night	0	0	33	101	50	22	206
Total	3	283	431	665	155	53	1590
Total Tested	3	283	431	665	155	53	1590
**2.2 Ardenne ** **478 km²**	**18 September**	**October–** **18 December**	**January–** **19 March**	**April–** **19 December**	**20 January–** **20 November**	**20 December–** **21 March**	
Period	1	2	3	4	5	6	Total
N Days	17	91	89	274	334	120	925
ASF Fences (km)			36	85.3			121
Fenced Highway (km)	35.7						35.7
Additional ASF Fences in the Surrounding (km)				40			40
Infected Zone (164 km²)							
Area Searched (km²)	3.1	57.8	88.3	200.3	125.2	19.4	494.2
Time for Searching (man × hour)	28	468	2197	5104	3062	389	11,248
Found Dead	1 (0)	5 (0)	23 (16)	57 (30)	40 (1)	0	126 (47)
Sanitary Shots				2 (0)			2 (0)
Road-Rail Casualty			1 (0)	3 (0)	2 (0)		6 (0)
Hunted		380 (0)	52 (0)	29 (0)	4 (0)	4 (0)	469 (0)
Trapped				95 (2)			95 (2)
Shot at Night				20 (0)	17 (0)	4 (0)	41 (0)
Total	1	385	76	206	63	8	739
Total Tested	1 (0)	37 (0)	57 (16)	206 (32)	63 (1)	8 (0)	372 (49)
Incidence	0.00	0.00	0.21	0.16	0.01	0.00	0.07
White Zone 314km²							
Area Searched (km²)	0.0	66.4	34.5	43.2	23.9	0.0	168.0
Time for Searching (man × hour)	0	354	348	1463	699	0	2864
Found Dead	2	7	9	21	40	1	80
Sanitary Shots	2		4	3	1		10
Road-Rail Casualty	1	2	6	18	6	4	37
Hunted	0	868	185	875	126	65	2119
Trapped				598	147	22	767
Shot at Night				22	194	26	242
Total	5	877	204	1537	514	118	3255
Total Tested	5	43	134	1163	454	84	1883

**Table 3 pathogens-12-00152-t003:** Evolution of the Belgian ASF management zoning area (km²) from September 2018 to March 2021.

Zone	ASF-Status	Period 1	Period 2	Period 3	Period 4	Period 5
**Gaume**	Infected km²	417	417	454	454	418
	White km²	211	211	173	173	210
**Ardenne**	Infected km²	0	0	142	164	164
	White km²	0	279	336	314	314

**Table 4 pathogens-12-00152-t004:** Chronology of restrictions applied in ASF-infected forest areas, based on successive government and ministerial regulations.

Period	Date	Measure
1	September 2018	Ban on feeding and hunting everywhere Ban on circulation in the forest (for tourist and forestry activities) Only active search/removal/analysis of carcasses by the authorities with biosecurity
2	October 2018	Authorisation to cull only for the authorities with appropriate means according to the epidemiological situation (e.g., trap building and management)
3	January 2019	Possible access for emergency and network maintenance services (GSM, railway,…), owners of houses enclosed in the forest, spruce bark beetle disease managers
	March 2019	Access only on paved roads to agricultural plots enclosed in the forest, for the recovery of cut logs stored at the side of the roads, for the census of wildlife by night counting, to sites of cultural interest
4	April 2019	Re-opening of some safe areas for walkers, only on marked trails and roads. The safe areas are regularly updated according to the epidemic situation
	June 2019	Access allowed for hunters for single hunting from a high seat on baiting point Temporary derogation for camp sites enclosed in the forest
	October 2019	Closing of the infected forest for the walkers to allow complete depopulation
5	January 2020	Authorisation for certain light work to maintain the forestry plantations in the infected forest under restrictive conditions (hand-held equipment only).
	May 2020	Authorisation for certain heavy forestry work including logging in the forest (except in wetlands) under restrictive conditions, namely training about biosecurity and an active search for carcasses by the authorities in the area affected by the work. Disinfection of operating equipment is the responsibility of the authorities. Reopening of paths for tourists, only during daylight hours.
	August 2020	Authorisation to take firewood out of the forest and to prepare the hunting season (wild boar, red, and roe deer), under restrictive conditions.
	September 2020	Driven hunting with limited number of hunting dogs allowed
6	November 2020	End of the restrictions during the day No disinfection required anymore Night access to the forest remains prohibited for safety reasons (night shooting)
	April 2021	Free access to the forest

**Table 5 pathogens-12-00152-t005:** According to the zone (Gaume vs. Ardenne, Infected vs. White), summary of the area covered (ha forest) and the time spent (h) by searching teams, the speed of searching (ha/h), and the number of times the area has been covered.

Zone	ASF-Status	Forest Area (ha)	Searching Effort (ha)	Searching Efforts (h)	Speed of Searching (ha/h)	Number of Times the Zone Has Been Covered	% Effort
Ardenne	Infected	9687	49,416	11,248	4.4	5.1	19
	White	17,624	16,803	2864	5.9	1.0	5
Gaume	Infected	21,721	199,807	41,883	4.8	9.2	69
	White	8222	55,526	4637	12.0	6.8	8
Total		57,254	321,552	60,631	5.3		100

**Table 6 pathogens-12-00152-t006:** Mortality data recorded during the last ASF-free hunting season in 2017–2018 according to infected, white zones and Wallonia, and mortalities recorded in 2018–2019, 2019–2020 and 2020–2021, the three years following the ASF outbreak in wild boar (+ evolution in % compared to 2017–2018).

ZONE	2017–2018	2018–2019	2019–2020	2020–2021
Infected zone (618 km²)	754	1505 (+100%)	607 (−19%)	120 (−84%)
White zone (488 km²)	1042	1803 (+73%)	2516 (+141%)	526 (−50%)
Wallonia (16901 km²)	29678	35488 (+20%)	36053 (+21%)	26686 (−10%)

## Data Availability

The data presented in this study are available on request from the corresponding author.

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
