# Peer review of "Management of a Focal Introduction of ASF Virus in Wild Boar: The Belgian Experience"

_pathogens, 2023, doi:10.3390/pathogens12020152_

Round 1

Reviewer 1 Report

The manuscript is a very valuable piece of work, detailing the measures and expansion of the ASF epidemic in Belgium. The authors describe this in a more narrative way, which is presumably the only possible way to describe it. However, some parts should be moved from the results section to the MM section, and should be described in more detail (Euclidian distance, fence effectiveness …)

The English should be checked, the use (or not use) of commas has to be implemented throughout the manuscript. I have only marked a few of them, there are many more instances where commas are missing.

Most of my comments are related to the use of English – however, the manuscript should be checked by a native speaker!

As a general comment: Abbreviations should be written out when they are used for the first time (even commonly known ones like GPS and GIS)

Page 1, line 30: before all successfully eradicated: do you mean were successfully eradicated?

Furthermore, I would prefer eliminated instead of eradication – as eradication means world-wide, while elimination means in a local area

Page 1, line 33: wild boar, (comma) endangering …

Page 1, line 44: no case has ever been detected in domestic pigs IN BELGIUM during that time (haven’t there been introductions in the 1980s into Belgium?

Page 3, line 75: the public forest in in the majority comprising – this sounds a bit awkward, maybe: the public forests comprise in the majority …

Page 3, line 106: for how long was the stand-still? 

Page 4, line 149: it is not clear what roles the hunters had – they were responsible for the depopulation, but not for the search?

Page 4, line 151: the authorities did the depopulation in the infected zone. But who of the authorities? Presumably also hunters? 

Page 4, line 151: when were driven hunts allowed again? 

Page 4, line 155: mortality report sounds not exactly what it is, the wild boar were shot, thus not sure I would call this mortality.

Page 4, line 156: for how many percent of the culled wild boar was the centroid used? 

Page 5, line 176: please write the name of the reference [8]

Page 5, line 180: you can omit the before carcasses

Page 5, line 184: people involved in the carcass search – who where they, were they volunteers, or forestry persons? How many in total?

Page 5, line 186: hunters HAD been trained

Page 5, line 197: infected polygons: how were they calculated? Convex? Concave? Which package in R? 

Page 6, line 211: I would rather write: elimination than extinction – you do not know if it was extinct from that area, you just did not find it anymore, and extinction would mean world-wide

Page 6, line 213: what means very low? 

Page 6, line 215: table 2: until (instead of till)

Page 6, line 220: table 2: the heading with the dates is hard to read, would be better to use two lines for this (like Dec20- Mar21)

Page 8, line 222: the method with the Euclidean distances should be described in the MM section. Furthermore, does the vector not show from (instead of to) the infected area of the previous month?

Page 8, line 225: figure 3: there are only three colors in the graph, but you have four different descriptors: (1) ASF positive (2) culled (3) found dead bones (4) found dead fresh

Furthermore, it would be good to write just found dead bones and found dead fresh (rather than using the underscore: found_dead_bones) – presumably you can omit (1) ASF positive, as it is written in the Y-axis reference

Does fresh carcasses refer to found_dead_fresh? Would be good to use the same term.

Page 8, line 232: what are external plots? 

Page 9, line 245: when were these measures implemented to human activity, in which period? 

Page 12, line 257: black circles rather than grey? 

Page 12, line 258: areas searched at least once DURING THE PERIOD (please add this, if this is meant)

Page 12, line 259: please use until instead of till

Page 13, line 262: Gaume is above and Aredenne below

Page 13, line 263: please use until instead of till

Page 14, line 271: should this read man*hour (or man-hour – as written below) (because it is not 53000 man per hour)

Page 15, line 276: please omit the before Gaume

Page 16, line 284: figure 6: why calculated? I would assume the time has been measured and recorded, not calculated? 

Page 16, line 268: please use until instead of till

Page 16, figure 6: I would switch the panels of the figure, and reorder them according to figure 5, meaning having Gaume in the top two panels and Ardenne in the lower two panels.

Page 16, line 288: I would write fences instead of fencing

Page 16, line 294: confining virus circulation: fences won’t stop the virus, they can only stop the wild boar – you might want to rewrite this phrase 

Page 16, lines 294 ff: how often was this done before an animal was found positive outside of the fences? 

Page 17, figure 7: why use the same color for the ZI and ZNI and then distinguish it in the legend of the figure? Maybe use different colors or don’t distinguish it in the legend

What is the meaning of N801 and N891? 

Page 17, line 301: it is not clear where those 10 fences segments are in the figure? If I try to count them, I find more or less than 10 (depending on the color)

Page 17, lines 303ff: this section is not clear yet, and should be described in more detail in MM. could you depicture these 10 segments in a map? 

Page 17, line 320/320: why is there still doubt, what way? 

Page 17, line 321: virologically positive (rather than viro-positive)

Page 18, figure 8: presumably the dotted line is the fence and the negative numbers are inside the infected area and the positive numbers outside the infected area? This should be described in more detail in the description of the figure (lines 327/328)

Page 18, line 329: what methods of spatial temporal analysis has been employed? Please describe in the MM section

Page 20, figure 10: maybe show it similar to Figure 5 (Gaume in the two top panels and Ardenne in the two lower panels)

Please describe in the description of this figure (line 363) that the lines denote the periods. Furthermore, I only count 5 periods, not 6 in the figure

Page 21, line 380: elimination in Belgium instead of eradication (eradication would be nice, but not achievable in the moment)

Page 21, line 382: experts’ (delete the space after expert) – is there a reference for this experts’ report? 

Page 21, line 386: large instead of high

Page 21, line 391: applying the same (omit ‘on’)

Page 21, lines 401: repetition ‘in the heart of the forest’: the two sentences seem to repeat each other

Page 21, lines 405/406: in September the maize was already harvested? This seems unlikely? 

Page 22, lines 407: change the full stop into a comma, after the reference [14]

Page 22, line 416: please write the name of the author of reference [18]

Page 22, line 423: at the end of the bracket of the references, there seems to be a surplus bracket

Page 22, lines 423/224: the location of the carcasses on them own, does not allow fully to describe the epidemic phase. First of all, you do not know if you found all the carcasses and secondly, you would have to take into account the time when the animals died (which does not mean detected; detection could me much later than death)

Page 22, line 435: why not write GIS – you used that abbreviation already before (without writing the full name – would make more sense to write the full name in the first instance and thereafter use the abbreviation)

Page 22, line 442: what are outer ASF-positive wild boar? 

Page 22, line 444: had been noticed (instead of have been noticed)

Page 22, line 451: searching of carcass searches – seems to be repetition. Furthermore, this sentence does not make sense. What interventions of the forestry persons? 

Page 22, line 459: Unmanned aerial vehicles (why capitals?)

Page 23, line 468: were found, instead of were to be found

Page 23, line 469: the time spent searching WAS (instead of is)

Page 23, line 470: WAS (instead of is)

Page 23, line 487: fences will not stop the virus’s spread, only the movement of wild boar

Page 23, line 491: better use area instead of surfaces

Page 23, line 494: no optimal size of fenced area – how do you know, there was no optimal size? 

Page 23, line 495: all requirements (plural please)

Page 23, line 510: EFSA reference has no number formatting

Page 24, line 514: IN depopulation instead of to

Page 24, line 527: please state the name of the author of the reference

Page 24, line 541: elimination not extinction, furthermore, do you really mean rate? Rate is per time interval.

Page 24, line 561: please state the name of the author of the reference

Page 25, line 564: eliminate instead of eradicate

Page 25, line 566: please state the name of the author of the reference

Author Response

Dear Reviewer,

many thanks for your hard work that really improved our manuscript.

Kind regards

Reviewer 2 Report

The Topic of the Article is "Management of a focal introduction of ASF virus in wildlife: the Belgian experience", however, "wildlife" is a very wide definition. African swine fever (ASF) affects only suids - domestic and wild and in Belgium ASF was introduced into the wild boar population, that wise for more clarity it would be suggested to amend the Topic and instead of "wildlife" use "wild boar".

Lines 11-12 - it is suggested to use instead of "swine" the wording "suids" and to clarify - instead of "wild" use "wild boar".

Line 13 - instead of "virus" use "African swine fever virus" which provides more clarity in the Abstract.

Lines 29-34 - it would be necessary to mention, that several incursions of the ASF virus before 1995 were related to the ASF genotype I virus strain, however, the introduction of the ASF virus in Georgia in 2007 was related to the ASF genotype II virus strain.

Line 35 - instead of "wildlife" use "wild boar".

Line 46 - the OIE has officially changed its name to WOAH (World Organisation for Animal Health) and that abbreviation should be used. 

Line 62 - 63 - Please clarify, whether "first EU disease control zoning" refers to ASF, as well as whether "virus" refers to "ASF virus".

Line 85 - Please clarify, whether the "outbreak" refers to "ASF outbreak in wild boar". If so, an abbreviation with the explanation would be useful to be used all over the Article.

Line 137 - it is suggested to use the term "movement" instead of "migration" -  "to avoid wild boar movement".

Line 247 - please explain the meaning of the abbreviation "EuVet".

Line 345 - the topic "Depopulation" should be reconsidered or renamed, while depopulation means the killing of all animals in the area, however, the authors state, that the remaining population was estimated, which means, that the depopulation was not successful. It would be better to change the wording and instead the term "depopulation" to use "reduction of the wild boar population" or similar and to use the same term everywhere in the Article. 

Line 383 - abbreviation "EFSA" should be explained.

Lines 385-386 - Part II at that time in 2018 has been set by the Commission Implementing Decision 2014/709, but not in terms of EC Directive.

Author Response

Dear Reviewer,

thank you for your comments that greatly improved our manuscript

Kind regards
